# Phasor Wave-Field Simulation Providing Direct Access to Instantaneous Frequency: A Demonstration for a Damped Elastic Wave Simulation

**René Hammer** *  , **Lisa Mitterhuber**  and **Roland Brunner**

Materials Center Leoben Forschung GmbH, Roseggerstrasse 12, 8700 Leoben, Austria;
lisa.mitterhuber@mcl.at (L.M.); roland.brunner@mcl.at (R.B.)
* Correspondence: rene.hammer@mcl.at

**Abstract:** In this work, we describe and simulate a wave field as a phasor field by simultaneously propagating its real and imaginary parts. In this way, the unique phase angle is directly available, and its time derivative determines the instantaneous frequency. We utilize the concept to describe damping in elastic wave propagation, which is of high importance in several engineering and research disciplines, ranging from earth science and medical diagnosis to physics.

**Keywords:** elastic waves; ultrasound; lossy media; damping; phasor field; instantaneous frequency

## 1. Introduction

Elastodynamic wave propagation [1] is of crucial importance on a broad spectrum of length scales: starting in the kilometer range, from earth science including earthquake research and geophysical exploration [2–4], going down the scale to the millimeter and sub-millimeter range in medical diagnosis and therapy [5,6], sensors and actuators [7–9], material science [10–12], product and process monitoring [13–15], approaching the smallest scale, in non-destructive testing of microelectronic and nanoelectronic components [16,17] phononic crystals and hypersound propagation [18]. In many cases, damping of elastic waves is significant and desired; hence, it must be considered in the simulation of wave propagation phenomena [19–21]. For this task, linear viscoelastic material behavior is commonly assumed, which is influenced by the entire material loading time history. This behavior corresponds to a numerically expensive convolution of the change of stress/strain, with a creep/damping kernel, respectively. Consequently, damping was already introduced into time-domain viscous wave simulation, usually approximatively aiming to reduce numerical cost. This was done by direct convolutional models to intermediate Maxwell and Zener body models [22], towards spatially coarse sampled single parameter methods [23]. Excellent literature surveys on this topic are available, and we suggest the review of Moczo et al. [24].

Convolution in the time domain corresponds to a simple multiplication in the frequency domain. In this short communication, we compactly present a combination of three general concepts to transfer this efficiently to the time domain. The first concept is the representation of a wave by local amplitude and phase (phasor). From the practical perspective, one can choose an arbitrary time-domain wave propagation algorithm to propagate the real part and the imaginary part of the phasor simultaneously. The second concept is the instantaneous frequency, defined as the rate of change in time of the phase [25,26]. The third concept uses the instantaneous frequency to implement an arbitrarily frequency-dependent attenuation of an elastic wave packet. An arbitrary time-stepping wave propagation algorithm can be used if for example, after every propagation time-step, a damping time-step is included. To demonstrate the methodology, we provide a practical example of damped elastic wave propagation using the velocity-stress formalism in the form of the elastic finite integration technique (EFIT) [27].

The paper is structured as follows: in Section 2 we provide the basic concepts (phasor, instantaneous frequency, and wave attenuation); in Section 3 we introduce the use case of elastic wave propagation in solid media; in Section 4 we provide results for damped elastic wave packet dynamics; and in Section 5 we conclude and provide an outlook for potential future developments.

## 2. Materials and Methods

### 2.1. The Phasor Concept

The classic wave theory assumes that a "slowly" varying function exists so that the wave-field, which is the particle deflection $u$ at a certain time $t$ at a certain position $x$ in space, can be written as

$$u(x,t) = Re\left[a(x,t)e^{i\cdot\theta(x,t)}\right] \tag{1}$$

with the time-varying amplitude $a(x,t)$, the phase-angle $\theta(x,t)$, where $i$ is the imaginary unit. Commonly the real part of the complex-valued function is taken because only real values can be measured directly in nature. The complex representation using the length of a pointer and a phase-angle called a phasor, evolving in time, is shown in Figure 1a. The complex wave-field as represented by a phasor in each point in space is shown in Figure 2b.

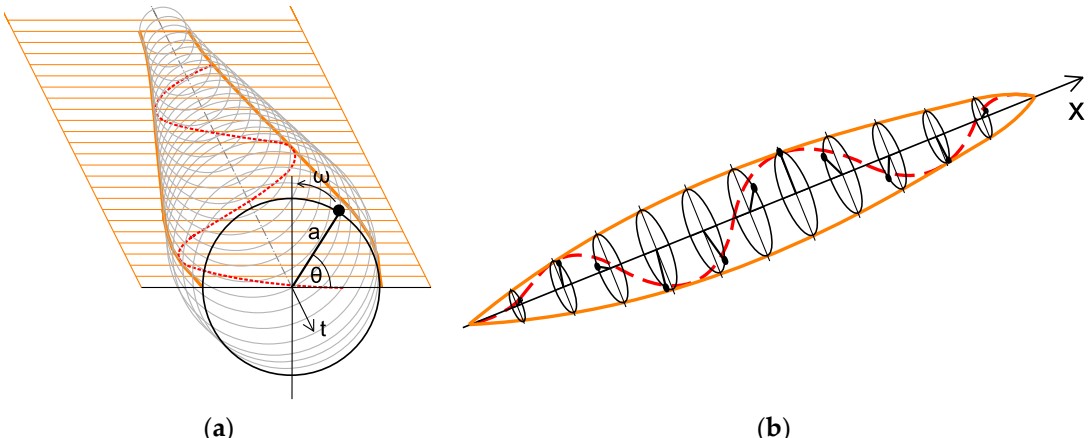

|     |     |
| :-: | :-: |
| (**a**) | (**b**) |

**Figure 1.** (**a**) The phasor concept: A phasor is uniquely defined by its amplitude $A$ and its phase angle $\theta$. (**b**) The phasor field is defined as having a phasor in each point in space.

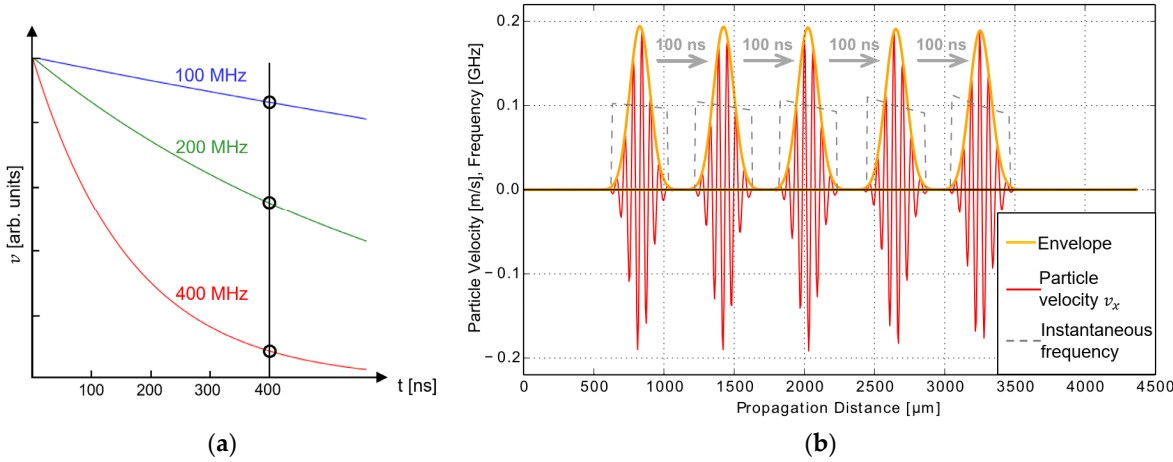

|     |     |
| :-: | :-: |
| (**a**) | (**b**) |

**Figure 2.** (**a**) The frequency estimator-based implementation of frequency-dependent damping shown for pressure wave propagation in water. (**b**) Propagation of a 100 MHz ultrasound pressure wave packet in aluminum. The instantaneous frequency computation reveals the numerical dispersion because a coarse spatial grid (10 grid points per wavelength) is chosen intentionally.

### 2.2. The Instantaneous Frequency Concept

The instantaneous frequency (angular speed) $\omega$ is defined as the time derivative of the phase angle

$$\omega = \frac{d\theta}{dt} . \tag{2}$$

As a remark, the angular speed is defined in a natural way by the change of the phase angle in time, similarly to the fact that the linear speed (velocity) is the change of the spatial coordinate in time.

In some cases, the instantaneous frequency can take on negative values and special care has to be taken to treat such cases properly [28]. However, in the examples, shown below, we deal with signals which are locally symmetric to the zero mean, which show strictly positive instantaneous frequency [29].

### 2.3. The Wave Attenuation Concept

When a wave propagates in space, the rate of reduction in amplitude is proportional to the amplitude itself

$$\frac{da(x)}{dx} = -\alpha(\omega)a(x) \tag{3}$$

with the damping factor $\alpha$ being a function of the frequency. This leads to the exponential law for the amplitude evolution of a damped wave traveling a certain distance $\Delta x$

$$a(x + \Delta x) = a(x)e^{-\alpha(\omega)\,\Delta x} . \tag{4}$$

The damping coefficient $\alpha(\omega)$ can be an arbitrary function of frequency $\omega$ in general. An analogous equation holds for the evolution of a wave in time

$$a(t + \Delta t) = a(t)e^{-\delta(\omega)\,\Delta t} \quad \text{with } \delta(\omega) = v_p\alpha(\omega) \tag{5}$$

where $v_p$ stands for the phase velocity.

### 3. The Use Case, Elastic Wave Propagation

We demonstrate the usage of the concept of instantaneous frequency for modeling damping for elastic wave propagation. An arbitrary method for the propagation of the elastic wave in space and time can be used. In this work, we apply the velocity-stress representation of the elastodynamic equations in the form of the EFIT discretization [3]. The field variables are the particle velocity $\vec{v}$, being a vector in three dimensions, and the symmetrical Cauchy stress tensor $\boldsymbol{\sigma}$ in 3D having up to six independent components (3 components in 2D), respectively.

The ingredients for describing elastic wave propagation are:

- The kinetics which has its origin in Newton's second law and is also called the Cauchy momentum equation $\rho\frac{\partial \vec{v}}{\partial t} = \vec{\nabla} \cdot \boldsymbol{\sigma} + \vec{f}$, where $\vec{v}$ the particle velocity field is the first derivative of the displacement field $\vec{u}$ with respect to time, $\rho$ stands for the mass density, and $\vec{f}$ represents the body forces (like gravity, electro-magnetic forces, etc.);
- The kinematics, relating the velocities of a material point to the strain rates $\frac{d\boldsymbol{\epsilon}}{dt} \approx \frac{1}{2}\left[\left(\vec{\nabla} \otimes \vec{v}\right) + \left(\vec{\nabla} \otimes \vec{v}\right)^T\right]$;
- The material law relating stress and strain $\boldsymbol{\sigma} \approx \lambda\,tr(\boldsymbol{\epsilon}) + 2\mu\,\boldsymbol{\epsilon}$, here shown for an isotropic material with $\lambda$ and $\mu$ being the first and second Lame constants representing the stiffness of the material.

In this framework, a general viscoelastic response can be obtained by introducing particle velocity proportional (dissipation, analog Maxwell body) $\vec{v}(t + \Delta t) = A(\omega)\vec{v}(t)$ damping in combination with a stress proportional (stress relaxation, Kelvin-Voigt type)

$\sigma(t + \Delta t) = B(\omega)\sigma(t)$ relaxation mechanism. The detailed equations for the velocity-stress formalism are provided in the Appendix A. All initial waves and wave packets are initialized with a reference phase (real part of the phasor) and in parallel with a 90° phase shift (imaginary part of the phasor). As a remark, if only the real part of the signal is known, then the imaginary part is obtained by the Hilbert transform [29]. This can in principle also be done at every timestep, but because of the non-locality of the Hilbert transform in time, this is numerically costly. Therefore, we choose to propagate the real and the imaginary part, avoiding the use of the Hilbert transform. At ultrasound frequencies, velocity proportional damping, which scales with the square of the frequency is frequently the dominating damping mechanism. Examples for this behavior are liquids (thermoelastic relaxation, shear, and bulk viscosity) and many solid materials (thermoelastic relaxation, phonon-phonon, and electron-phonon interaction) [10]. Therefore, the following expression is used in all following examples:

$$A(\omega) = e^{-\beta \, \omega^2 t} \tag{6}$$

## 4. Results

First, the performance of the instantaneous frequency-based implementation of frequency-dependent damping is validated. For this task, the damping is evaluated by investigating the time evolution of the energy within the simulation domain (periodic boundary conditions are chosen). The total energy is given by the kinetic energy plus the potential energy, and each on time average is $\frac{1}{2}$ of the total energy (remark: without implementing damping, the EFIT scheme is strictly an energy preserving one [27,30]). Therefore, the average velocity is proportional to the square root of total kinetic energy. Comparing the numerical results, after 400 ns with the values from the analytical formula Equation (6) with a plane wave of frequency ($f = \omega/2\pi$) 100 MHz, 200 MHz, and 400 MHz, gives a relative error for the mean velocity $[(v_{num} - v_{ana})/v_{ana}]$ of $4 \times 10^{-6}$, $16 \times 10^{-6}$ and $65 \times 10^{-6}$, respectively. The decay of the mean velocity amplitude is shown in Figure 2a.

The propagation of ultrasound in aluminum (the material properties are provided in Table 1) is shown in Figure 2b The initial condition is a modulated Gaussian-shaped pressure wave-packet with a mean frequency of $f$ = 100 MHz and a pulse-width (two times Gaussian sigma) of 40 ns. The pressure wave velocity is $c_p = [(\lambda + 2\mu)/\rho]^{1/2} \approx 6316$ m/s and the corresponding wavelength is $l_p = c_p/\nu \approx 63$ μm. We choose a coarse grid of 10 grid points per wavelength to reveal the numerical dispersion and the possibility of resolving this effect by computing the instantaneous frequency. As is expected, the higher frequency components propagate slower on the grid, decreasing the mean instantaneous frequency content at the front and increasing it on the rear of the wave packet, respectively.

**Table 1.** The model material properties for the elastic wave propagation examples.

| Material | Mass Density | 1st Lame Const. | 2nd Lame Const | Damping [1] |
|---|---|---|---|---|
| | $\rho$ [kg/m$^3$] | $\lambda$ [GPa] | $\mu$ [GPa] | $\beta$ [Neper/(s Hz$^2$)] |
| Aluminum | 2700 | 55.5 | 26.1 | - |
| Water | 1000 | 2.08 | 0 | $3.7 \times 10^{-11}$ |

[1] The prefactor for damping coefficient scaling with frequency squared, from [31]. The value in aluminum is much lower than water and is therefore neglected.

The Figure 3 shows a chirped signal, linearly ramped from 20 to 120 MHz (60 MHz at the center of the Gaussian, everything outside a 2 sigmas Gaussian width is set to be equal to zero) with a pulse width of 20 ns. A gap of 100 μm, between two aluminum bodies filled with water is introduced to form a cavity. A "ringing" behavior is observed where the wave packet trapped in the cavity reflects back and forth and "radiates" into the aluminum bodies. Because this setup does not introduce a significant frequency filtering effect, the "radiated" wave packets show a similar instantaneous frequency profile compared to the initial wave packet.

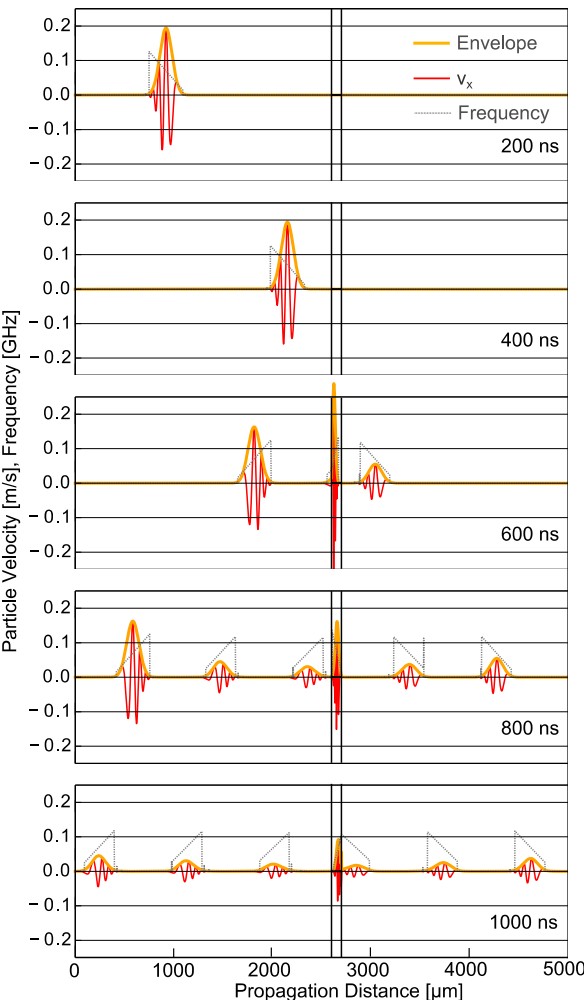

**Figure 3.** The ringing effect of a chirped ultrasound wave packet (low frequencies in the front of the packet) in aluminum approaching a gap (cavity) of 100 μm filled with water (indicated by the vertical parallel lines). Subsequent time-steps are shown at 200 ns and 400 ns before the wave packet reaches the cavity and for 600 ns, 800 ns, and 1000 ns, while the wave packet interacts with the cavity. The instantaneous frequency profiles are shown as gray dotted lines.

## 5. Conclusions and Outlook

In this short communication, we demonstrate that numerical propagation of the phasor field provides the instantaneous frequency at each point in space and time. We have demonstrated this for the case of wave packets, where a dominant center frequency is present and was subsequently revealed as the instantaneous frequency. This enables a conceptually simple and numerically cheap implementation of arbitrarily frequency-dependent damping behavior. It is the topic of future studies to investigate the behavior for cases where multiple frequencies are present simultaneously in the same point in space. One option is to discretize the frequency space into several frequency windows, subsequently translate them to wave number windows using the known dispersion relation, and extract the signal for each desired wave number window by applying a filter in space. Another idea is to use the concept to improve the efficiency of numerical schemes by repairing the numerical dispersion error (=wrong phase velocity for high-frequency content/coarse grid), by an instantaneous frequency informed tuning of the local velocity. This would allow a lower number of grid points per wavelength for a targeted accuracy. Last but not least, the instantaneous frequency is also defined for nonlinear wave propagation, which could open new possibilities in this field.

**Author Contributions:** Conceptualization, R.H. and L.M.; methodology, R.H.; software, R.H.; validation, R.H.; investigation, L.M.; resources, R.B.; writing—original draft preparation, R.H.; writing—review and editing, L.M. and R.B.; visualization, R.H.; supervision, R.B.; project administration, R.H. and R.B.; funding acquisition, R.H. and R.B. All authors have read and agreed to the published version of the manuscript.

**Funding:** The authors gratefully acknowledge the financial support under the scope of the COMET program within the K2 Center Integrated Computational Material, Process and Product Engineering (IC-MPPE) (Project No 859480). This program is supported by the Austrian Federal Ministries for Climate Action, Environment, Energy, Mobility, Innovation, and Technology (BMK) and for Digital and Economic Affairs (BMDW), represented by the Austrian research funding association (FFG), and the federal states of Styria, Upper Austria, and Tyrol.

**Data Availability Statement:** Data available on request from the authors.

**Conflicts of Interest:** The authors declare no conflict of interest.

## Appendix A

For the numerical simulation of the elastic wave propagation, we use velocity-stress formalism on a staggered grid, also known as the elastic finite integration technique (EFIT) [1]. For compactness, the algorithm is provided briefly here, and all simulations are performed in 2D. The numerical propagation scheme in time, with the discrete-time index n for one unit cell in time, is illustrated in Figure A1a. As a remark, the velocity is shifted by $(\Delta t/2, \Delta x/2)$ relative to the stress values within the unit cell. The reason for this is to allow a central finite difference approximation for the time and spatial derivatives. As a general initial condition (index $n = 0$), the stress and the particle velocity in the $t^0$ time cell are given. In a leap-frog manner in the first partial step, the new stress (at $t^{n+1}$) is computed from the old stress and velocity values (at $t^n$). The new velocity values are computed by the now known new stress (at $t^{n+1}$) and old velocity values (at $t^n$). The discretization of the field variables on the staggered spatial grid is shown in Figure A1. This, together with the time staggering, has the purpose of significantly reducing the numerical dispersion errors [30]. The material parameters, mass density $\rho$, and the stiffness Lame constants $\lambda$, $\mu$ have to be averaged on the integration cells, as shown in Figure A1b [27].

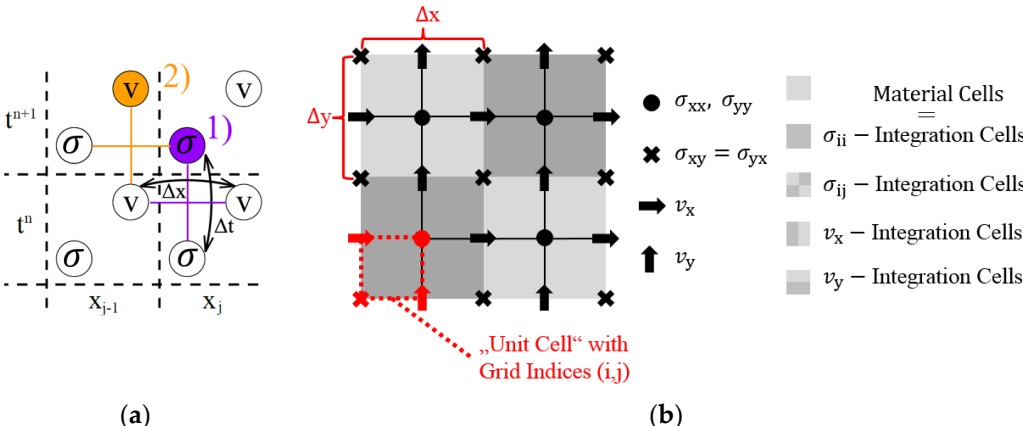

**(a)**  **(b)**

**Figure A1.** (**a**) The time integration within the velocity-stress EFIT scheme is done by staggering in time, called a leap-frog scheme. (**b**) Spatial grid by staggering of the field variables and integration cells.

General damping behavior is implemented, introducing frequency-dependent multiplicative factors, including the stress relaxation (via the 3 matrix components $\gamma_{xx}(\omega)$, $\gamma_{xx}(\omega)$, $\gamma_{xx}(\omega)$, and velocity proportional to the damping:

$$\sigma_{xx}^{i,j,n+1} = e^{-\gamma_{xx}(\omega)\,\Delta t}\,\sigma_{xx}^{i,j,n} + \left(\lambda^{i,j} + 2\,\mu^{i,j}\right)\frac{v_x^{i+1,j,n} - v_x^{i,j,n}}{\Delta x}\cdot\Delta t + \lambda^{i,j}\frac{v_y^{i,j+1,n} - v_y^{i,j,n}}{\Delta y}\cdot\Delta t$$

$$\sigma_{xx}^{i,j,n+1} = e^{-\gamma_{xx}(\omega)\,\Delta t}\,\sigma_{xx}^{i,j,n} + \left(\lambda^{i,j} + 2\,\mu^{i,j}\right)\frac{v_x^{i+1,j,n} - v_x^{i,j,n}}{\Delta x}\cdot\Delta t + \lambda^{i,j}\frac{v_y^{i,j+1,n} - v_y^{i,j,n}}{\Delta y}\cdot\Delta t$$

$$\sigma_{yy}^{i,j,n+1} = e^{-\gamma_{yy}(\omega)\,\Delta t}\,\sigma_{yy}^{i,j,n} + \left(\lambda^{i,j} + 2\,\mu^{i,j}\right)\frac{v_y^{i,j+1,n} - v_y^{i,j,n}}{\Delta x}\cdot\Delta t + \lambda^{i,j}\frac{v_x^{i+1,j,n} - v_x^{i,j,n}}{\Delta y}\cdot\Delta t$$

$$\sigma_{xy}^{i,j,n+1} = e^{-\gamma_{xy}(\omega)\,\Delta t}\,\sigma_{xy}^{i,j,n} + \frac{4\Delta t}{\frac{1}{\mu^{i-1,j-1}} + \frac{1}{\mu^{i-1,j}} + \frac{1}{\mu^{i,j-1}} + \frac{1}{\mu^{i,j}}}\left[\frac{v_y^{i,j,n} - v_y^{i-1,j,n}}{\Delta x} + \frac{v_x^{i,j,n} - v_x^{i,j-1,n}}{\Delta y}\right]$$

$$v_x^{i,j,n+1} = e^{-\beta(\omega)\,\Delta t}\,v_x^{i,j,n} + \frac{2\Delta t}{\left(\rho^{i-1,j} + \rho^{i,j}\right)}\left[\frac{\sigma_{xx}^{i,j,n+1} - \sigma_{xx}^{i-1,j,n+1}}{\Delta x} + \frac{\sigma_{xy}^{i,j+1,n+1} - \sigma_{xy}^{i,j,n+1}}{\Delta y}\right]$$

$$v_y^{i,j,n+1} = e^{-\beta(\omega)\,\Delta t}\,v_y^{i,j,n} + \frac{2\Delta t}{\left(\rho^{i,j-1} + \rho^{i,j}\right)}\left[\frac{\sigma_{yy}^{i,j,n+1} - \sigma_{yy}^{i,j-1,n+1}}{\Delta y} + \frac{\sigma_{xy}^{i+1,j,n+1} - \sigma_{xy}^{i,j,n+1}}{\Delta x}\right]$$

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
