# Peer review of "Phasor Wave-Field Simulation Providing Direct Access to Instantaneous Frequency: A Demonstration for a Damped Elastic Wave Simulation"

_acoustics, doi:10.3390/acoustics3030032_

Round 1

Reviewer 1 Report

With the concept of instantaneous frequency, the frequency-dependent attenuation factor was introduced during sound wave propagation, and simulation was conducted.

1、There should be some restrictions in the application of instantaneous frequency, otherwise physical contradictions could occur. For example, when processing wideband signals, the instantaneous frequency will be a negative frequency that is physically unexplainable. Also, there are often situations that the instantaneous frequency exceeds the range of the original signal. How did the author deal with these problems? Besides, in the conclusion part, it was mentioned that “The instantaneous frequency computation from the phasor field is applicable for the case when different frequency components are separated in space. The example of elastic wave packet dynamics fulfills this requirement in most cases.”, which seems not to have a clear meaning. “Different frequency components are separated in space” should not be the characteristics of (wideband) pulse signals, while wideband pulse signals are often used in practice.

  1. It was mentioned in Sec. 2.3 that "All initial waves and wave packets are initialized with a reference phase (real part of the phasor) and in parallel with a 90° phase shift (imaginary part of the phasor)". That is to say, the imaginary part of the signal was obtained by 90° phase shift and parallel calculation of the excitation signal (real part of the phasor). So why not directly use Hilbert transform to obtain the imaginary part of the phasor after phase shift?
  2. In the introduction part, there is lack of literatures that directly use linear viscoelastic material on studying of sound attenuation propagation.
  3. About Eq. (5): ?(?)=???(?), what is the reason for using ?? (group velocity) here? And why not use the phase velocity?
  4. ?=?/2? (Sec. 4, Line 9), ? should be f.
  5. In Figure 2(b), through instantaneous frequency computation, the numerical dispersion was observed, which is difficult to avoid in numerical simulation. This leads to inaccuracy in calculation, thereby affecting the frequency-dependent attenuation factor. Is there a solution about this?

Author Response

We thank the reviewer for his insightful comments and for helping to improve the manuscript. We gratefully consider the suggestions and highlight the changes together with some additional corrections and suggestions from other reviewers in the manuscript (in red). 

Reviewer comment 1: “There should be some restrictions in the application of instantaneous frequency, otherwise physical contradictions could occur. For example, when processing wideband signals, the instantaneous frequency will be a negative frequency that is physically unexplainable. Also, there are often situations that the instantaneous frequency exceeds the range of the original signal. How did the author deal with these problems? Besides, in the conclusion part, it was mentioned that “The instantaneous frequency computation from the phasor field is applicable for the case when different frequency components are separated in space. The example of elastic wave packet dynamics fulfills this requirement in most cases.”, which seems not to have a clear meaning. “Different frequency components are separated in space” should not be the characteristics of (wideband) pulse signals, while wideband pulse signals are often used in practice.”

Response to RC1: We thank the reviewer for this insightful comment. Indeed, we forgot to mention this important point in the manuscript. We write now: “In some cases the instantaneous frequency can take on negative values and special care has to be taken to treat such cases properly. In the examples, shown below, we deal with signals which are locally symmetric to the zero mean, which show strictly positive instantaneous frequency [28].” We agree that some parts of the conclusion are misleading. We replace the mentioned critical parts by the sentence: “We have demonstrated this for the case of wave packets, where a dominant center frequency is present in the initial conditions and was subsequently revealed as the instantaneous frequency [29].”

Reviewer comment 2: It was mentioned in Sec. 2.3 that "All initial waves and wave packets are initialized with a reference phase (real part of the phasor) and in parallel with a 90° phase shift (imaginary part of the phasor)". That is to say, the imaginary part of the signal was obtained by 90° phase shift and parallel calculation of the excitation signal (real part of the phasor). So why not directly use Hilbert transform to obtain the imaginary part of the phasor after phase shift?

Response to RC2: Regarding the Hilbert transform we add the comment: “Remark: If only the real part of the signal is known, the imaginary part is obtained by the Hilbert transform [29]. This can in principle be done also at every timestep, but because of the non-locality of the Hilbert transform in time this is numerically costly. Therefore, we choose to propagate the real and the imaginary part, avoiding the use of the Hilbert transform.”

Reviewer comment 3: In the introduction part, there is lack of literatures that directly use linear viscoelastic material on studying of sound attenuation propagation.

Response to RC3: As the manuscript was meant to be a “short communication”, and we agree that the literature list is not very comprehensive. We try to mitigate the mentioned specific weak point and add a few citations in the introduction to close the gap to recent works.

Reviewer comment 4: About Eq. (5): ?(?)=?_??(?), what is the reason for using ?_? (group velocity) here? And why not use the phase velocity?

Response to RC4: There is no reason for this. This is an error in the manuscript. Thank you for pointing it out. We correct it accordingly to the phase velocity.

Reviewer comment 5: ?=?/2? (Sec. 4, Line 9), ? should be f.

Response to RC5: “\nu” for the frequency is fine in principle, but it can be confused with “v”. Therefore, we decide to follow the suggestion and use “f” instead.

Reviewer comment 6: In Figure 2(b), through instantaneous frequency computation, the numerical dispersion was observed, which is difficult to avoid in numerical simulation. This leads to inaccuracy in calculation, thereby affecting the frequency-dependent attenuation factor. Is there a solution about this?

Response to RC6: This is a very interesting question. The example mentioned by the reviewer was intentionally chosen (a very coarse grid) to show this effect. The pragmatic solution is to use a finer grid of course. But, as written in the outlook we hypothesize that the instantaneous frequency, in principle, can be used in combination with the known numerical-dispersion relation to correct the numerical error already in the propagation, e.g. by locally correcting the numerical velocity to obtain the desired one.

Reviewer 2 Report

A research article (manuscript ID: acoustics-1275407) entitled “Phasor Wave-Field Simulation Providing Direct Access to Instantaneous Frequency: Demonstration for Damped Elastic Wave Simulation” by a research team from Austria was submitted to the MDPI Acoustics Journal.

This paper has 9 pages including one table, 4 figures, and 27 references. In their short report, the authors describe and simulate a wave field as a phasor field when real and imaginary parts simultaneously propagate. They introduced a demonstration for damped elastic wave simulation.

The reviewer has looked through the paper and found that this presentation requires some corrections. For instance, the following corrections of the English language can be done:

1) Page 1, line 8 from the bottom: “represent” instead of “present”;

2) Page 3, the first line after equation (3): “the frequency” instead of “frequency”;

3) Page 4, the first line after figure 2: “are listed in Table 1) is shown” instead of “are in Table 1), is shown”;

4) Page 4, on the last two lines: “cp =((λ+2μ)/ρ)1/2 ≈ 6,316 m/s” instead of “cp =√((λ+2μ)/ρ )≈6316 m/s” and “lp = cp /ν ≈ 63” instead of “lp =cp /ν≈63”;

5) Page 5, line 7 above figure 3: “is set to be equal to zero)” instead of “is set to zero)”;

6) Page 6, in the context of the Appendix: please, use Italic for all the parameters such as n, t, x.;

7) Page 6, line 4th line from the bottom: There is no Figure 4!!!;

8) etc.

After equation (1) please also state that i is the imaginary unit.

The txt must be aligned with the page width but not to left that occurs after subsection 2.3 and above figure 3.

Please, use always “The” in any figure title after each figure number.

Figure 2. (a) The frequency” instead of “Figure 2. (a) Frequency”.

Figure 3. The ringing effect” instead of “Figure 3. Ringing effect”.

Table 1. The model” instead of “Table 1. Model”.

The English language must be polished. The titles of the corresponding figures must be improved. All the references should be in the MDPI format. Refs. [1,4,5,11,12,18,26] have no page numbers, for books and papers. It looks like book [1] has doubled title. Ref. [4] has no journal title. There are two authors for Ref. [11] such as D. Pandey and S. Pandey, for this book chapter. The paper requires a major revision.

Author Response

We like thank the reviewer for his/her careful reading and the valuable suggestions helping us to improve the manuscript. We gratefully consider the suggestions and highlight the changes together some additional corrections and suggestions from other reviewers in red. We checked and corrected the errors / missing info in the reference list.

Reviewer 3 Report

The manuscript addresses the usage of the concept of instantaneous frequency for modeling damping for elastic wave propagation. To do so, the authors apply the velocity-stress representation of the elastodynamic equations in the form of the elastic finite integration technique (EFIT) discretization.

The introduction provides a good overview of the state of the art and possible applications of the studied field. Also, the section corresponding to materials and methods clearly describes the most fundamental concepts that are necessary to understand the case of study (phasor, instantaneous frequency, and wave attenuation).

The methodology is well described and sufficient information is provided, both in the main body of the manuscript and the appendix, for the proper understanding of the content.

I have no comments to add that can increase the quality of the article, as all the contents are correctly introduced and developed, and the results and statements provided seem to be consistent. Nevertheless, a few formatting aspects must be considered, in the opinion of this reviewer, before the final publication of the manuscript:

  1. Introduction: In order to get a general overview of the issues addressed in the different sections of the manuscript, it would be interesting to write at the end of the introduction a small paragraph summarizing the structure of the manuscript and content of each section.
  2. Page 1, Paragraph 2, Line 4: There is a typo in the text between parenthesis, as it is written as “:=phasor”.
  3. There are two sections numbered as 2.3, which must be a formatting error. Section “2.3. The use case, elastic wave propagation” should be numbered as Section 3.
  4. Figure 2(b): The font size makes difficult to read the legend. Please, increase the size.

Author Response

We like thank the reviewer for his/her suggestions for improvement of the manuscript. We gratefully consider the suggestions and highlight the changes together some additional corrections and suggestions from other reviewers in red. At the end of the introduction we provide now a short paragraph regarding the structure of the manuscript.

Round 2

Reviewer 2 Report

A research article (a revised version of manuscript ID: acoustics-1275407) entitled “Phasor Wave-Field Simulation Providing Direct Access to Instantaneous Frequency: Demonstration for Damped Elastic Wave Simulation” by a research team from Austria was submitted to the MDPI Acoustics Journal.

This paper has 10 pages including one table, 4 figures, and 31 references. In their short report, the authors describe and simulate a wave field as a phasor field when real and imaginary parts simultaneously propagate. They introduced a demonstration for damped elastic wave simulation.

The authors have significantly improved their short report. As a result, the paper can be published.